# Development and Greenness Assessment of HPLC Method for Studying the Pharmacokinetics of Co-Administered Metformin and Papaya Extract

**DOI:** 10.3390/molecules27020375

**Published:** 2022-01-07

**Authors:** Mohamed A. Abdelgawad, Mohammed Elmowafy, Arafa Musa, Mohammad M. Al-Sanea, AbdElAziz A. Nayl, Mohammed M. Ghoneim, Yasmine M. Ahmed, Hossam M. Hassan, Asmaa M. AboulMagd, Heba F. Salem, Nada S. Abdelwahab

**Affiliations:** 1Department of Pharmaceutical Chemistry, College of Pharmacy, Jouf University, Sakaka 72341, Saudi Arabia; mhmdgwd@ju.edu.sa (M.A.A.); mmalsanea@ju.edu.sa (M.M.A.-S.); 2Department of Pharmaceutics, College of Pharmacy, Jouf University, Sakaka 72341, Saudi Arabia; melmowafy@ju.edu.sa; 3Department of Pharmacognosy, College of Pharmacy, Jouf University, Sakaka 72341, Saudi Arabia; akmusa@ju.edu.sa; 4Department of Chemistry, College of Science, Jouf University, Sakaka 72341, Saudi Arabia; aanayel@ju.edu.sa; 5Department of Pharmacy Practice, Collage of Pharmacy, AlMaarefa University, Ad Diriyah 13713, Saudi Arabia; mghoneim@mcst.edu.sa; 6Pharmacology & Toxicology Department, Faculty of Pharmacy, Nahda University in Beni-Suef (NUB), Beni-Suef 62521, Egypt; YasmineM.Ahmed@yahoo.com; 7Pharmacognosy Department, Faculty of Pharmacy, Beni-Suef University, Beni-Suef 62521, Egypt; hossam.mokhtar@nub.edu.eg; 8Pharmacognosy Department, Faculty of Pharmacy, Nahda University in Beni-Suef (NUB), Beni-Suef 62521, Egypt; 9Pharmaceutical Chemistry Department, Faculty of Pharmacy, Nahda University in Beni-Suef (NUB), Beni-Suef 62521, Egypt; asmaa.aboulmaged@nub.edu.eg; 10Pharmaceutics Department, Faculty of Pharmacy, Beni-Suef University, Beni-Suef 62521, Egypt; HebaF.Salem@yahoo.com; 11Pharmaceutical Analytical Chemistry Department, Faculty of Pharmacy, Beni-Suef University, Beni-Suef 62521, Egypt

**Keywords:** greenness assessment, RP-HPLC, metformin, pharmacokinetics, papaya

## Abstract

Foods with medical value have been proven to be beneficial, and they are extensively employed since they integrate two essential elements: food and medication. Accordingly, diabetic patients can benefit from papaya because the fruit is low in sugar and high in antioxidants. An RP-HPLC method was designed for studying the pharmacokinetics of metformin (MET) when concurrently administered with papaya extract. A mobile phase of 0.5 mM of KH_2_PO_4_ solution and methanol (65:35, *v*/*v*), pH = 5 ± 0.2 using aqueous phosphoric acid and NaOH, and guaifenesin (GUF) were used as an internal standard. To perform non-compartmental pharmacokinetic analysis, the Pharmacokinetic program (PK Solver) was used. The method’s greenness was analyzed using two tools: the Analytical GREEnness calculator and the RGB additive color model. Taking papaya with MET improved the rate of absorption substantially (time for reaching maximum concentration (T_max_) significantly decreased by 75% while maximum plasma concentration (C_max_) increased by 7.33%). The extent of absorption reduced by 22.90%. Furthermore, the amount of medication distributed increased (30.83 L for MET concurrently used with papaya extract versus 24.25 L for MET used alone) and the clearance rate rose by roughly 13.50%. The results of the greenness assessment indicated that the method is environmentally friendly. Taking papaya with MET changed the pharmacokinetics of the drug dramatically. Hence, this combination will be particularly effective in maintaining quick blood glucose control.

## 1. Introduction

Type 2 diabetes mellitus (T2DM) is a type of metabolic disorder that includes chronic hyperglycemia resulting from insulin insensitivity on appropriate tissues [1,2]. The two most important factors contributing to the development of T2DM are obesity and physical inactivity that lead to chronic hyperglycemia associated with microvascular and macrovascular complications resulting in severe damage to the vital organs [1,2,3]. Elevated blood glucose levels are usually accompanied by oxidative stress release that has a critical role in the destruction of pancreatic β-cells, excessive lipid peroxidation, and cellular damage. Current therapies implicate antioxidants as an adjuvant therapy for T2DM in order to decrease the impact of oxidative damage caused by unregulated glucose metabolism [4].

In the same vein, medicinal plants are progressively gaining global acceptability for their use as bioactive agents in pharmaceuticals. Some of the secondary consequences of diabetes, such as kidney damage, fatty liver, and oxidative stress, have been improved by new hypoglycemic medicines derived from plants [5,6]. Among those is *Carica papaya* that is a member of family *Caricaceae*, which is characterized by its edible, pleasant fruit that provides good nutritional value in addition to being easily digestible. It has been reported that papaya leaves can help with asthma, worming, and dysentery symptoms [7]. In addition, papaya leaf extract has been used as a treatment for cancer due to the presence of tocopherol, lycopene, flavonoid, and benzyl isothiocyanate. The antioxidant effects of papaya were found to be due to its soluble phenolic compounds as well as vitamin C [8,9,10]. A literature survey showed that fermented papaya preparation has the ability to reduce both basal and postprandial glycemia in-vitro and improves the lipid profile [11,12]. Moreover, it was reported that the hypoglycemic activity of papaya is thought to be due to vitamin C, fiber, flavonoid, and saponin [13].

The co-administration of herbal medicines with common antidiabetic drugs, like metformin (MET), on the other hand, has the potential to alter their pharmacokinetic and pharmacodynamic properties; hence, it was critical to look into herb–drug interactions. This study looked into the pharmacokinetics of MET and papaya extract when they are simultaneously administered.

Following a thorough study of the literature, several methods for analyzing MET alone in various samples (plasma, urine, liver, brain, kidney, and muscles) were reported, including HPLC [14,15,16,17,18,19], HILIC [20], LC/MS/MS [21,22,23,24], HILIC/MS [25], and GC/MS [26]. MET in conjunction with various medications in various plasma samples was also estimated by HPLC [27,28,29,30,31,32], LC/MS/MS [33,34,35,36,37,38,39,40], and HILIC/MS [41,42] methods.

Green chemistry principles have recently attracted scientists’ interest to reduce human health and environmental risks. Various criteria are currently utilized to evaluate the environmental impact of various analytical methodologies, however, multi-criteria decision analysis (MCDA) allows for a rating of available analytical methods based on many evaluation criteria at the same time. The Analytical GREEnness (AGREE) [43] calculator is a comprehensive, adaptable, and clear approach to assessing the greenness of analytical procedures. It is based on 12 green chemistry principles. The result is a colored pictogram that depicts the weak and strong points of the whole evaluated system. It is simple to carry out with the help of easy-to-use graphical user interface (GUI) software. Another tool for greenness assessment is the red, green, and blue (RGB) additive color model [44], which employs three basic colors to symbolize three important characteristics of the examined method: blue denotes productivity/practical efficacy, red represents analytical performance, and green signifies compliance with “green” chemical principles. The additive synthesis of the primary colors yields the method’s final color. The model also has a numerical parameter called “Method Brilliance,” and the evaluation is done with ordinary Excel worksheets.

In this study, our aim was to see if the pharmacokinetics of an oral hypoglycemic drug, MET, when given to rats as an animal model, are affected by taking papaya extract at the same time. After a few basic sample pretreatment steps, the samples were ready to be used and the proposed pharmacokinetic investigation was completed using a low-impact RP-HPLC method. In addition, method validation was carried out in accordance with Food and Drug Administration (FDA) [45] guidelines. 

## 2. Results and Discussion

### 2.1. Optimization of the Developed Method

Various aspects were investigated during method optimization so as to separate MET completely from the plasma matrix in a short time. Due to the high polarity of MET, it is difficult to keep it in the stationary phase. Hence, increasing its retention period was a challenge during the optimization step. Different stationary phases (given in experimental section) were tested. Unfortunately, there was no discernible difference in the separation effectiveness of these stationary phases.

The ZORBAX Eclipse Plus C_18_ (250 × 4.6 mm, 5 µm) (Santa Clara, CA, USA) column was then used to continue the optimization process. The previously published mobile-phase solvent mixture of acetonitrile (10 mM of KH_2_PO_4_ solution with varying ratios (from 60:40 to 40:60, *v*/*v*), and varying pH levels of the aqueous phase (from 3.8 to 7), were used to evaluate a number of mobile phase mixtures with varying flow rates (from 0.5 to 2 mL/min). All studies found that the plasma matrix impeded the detection of the early MET peak. All of these tests were subsequently performed again, but this time with methanol as an organic modifier, which is more environmentally friendly than acetonitrile. Unfortunately, this resulted in an unclear plasma matrix and MET signals. Water was then changed by 10 mM of KH_2_PO_4_ solution. The retention period of MET was greatly increased when water was used as the aqueous phase, however, extremely little retention was found when 10 mM of KH_2_PO_4_ solution was used. After that, lower concentrations of KH_2_PO_4_ solution (5 mM, 1mM, and 0.5 mM) were attempted with various methanol ratios. The plasma matrix and MET were completely separated based on the use of a mixture of 0.5 mM of KH_2_PO_4_ solution and methanol (65:35, *v*/*v*). Furthermore, the flow rate of the mobile phase was adjusted at different values (from 0.5 to 2 mL/min), where complete separation within an acceptable time occurred when using 1.7 mL/min. Scanning was done at various wavelengths (210, 225, 235, and 254 nm), with 235 nm providing the best sensitivity with the least amount of noise.

An internal standard is particularly valuable for improving the precision, accuracy, and resilience of a bio-analytical method throughout development. Additionally, it is vital to take variability in sample preparation and analysis into consideration, as well as to adjust for analyte loss during sample preparation. It was difficult to choose an internal standard that had the same chromatographic behavior as the analyte while also being completely separated from the analyte and plasma peaks. Additionally, it should have chemical and physical properties that are comparable to the target analyte in order to be extracted with the same method. There were tests on internal standards such as atenolol, metoclopramide, olanzapine, ibuprofen, and guaiphenesin (GUF). GUF was the best internal standard in terms of chromatographic performance in relation to the plasma matrix and MET separated peaks. Finally, within 7.5 min, a complete plasma matrix, MET, and GUF separation was accomplished and the resulted retention times were: 2.4, 4.9, and 6.3 min, respectively (Figure 1). In addition, the procedure for extracting MET from the gathered samples of plasma is crucial since it influences the approach’s sensitivity and specificity. Because it is a straightforward and economical procedure, the method of plasma protein precipitation was selected. Acetonitrile, ethanol, methanol, and aqueous perchloric acid were investigated. As a result of using methanol, the highest extraction recovery was obtained, as well as the largest protein precipitation (Table 1).

### 2.2. Method Validation

To ensure the bio-analytical approach established was valid, US Food and Drug Administration (FDA) [46] guidelines were followed. Three quality control samples (QCs) (low QC (LQC), middle QC (MQC), and high QC (HQC)) were prepared and utilized to assess the method’s validity.

The calibration standards and QC samples were prepared and used to validate the proposed method by diluting the previously prepared working solutions of MET and GUF with the mobile phase mixture to prepare their corresponding working solutions.

The calibration line was plotted at nine concentrations (0.5, 1, 3, 5, 10, 20, 25, 28, and 30 g/mL), and linearity, as measured by the correlation coefficient (***r***), was assessed using a least-squares regression line. The regression equation that was generated was found to be:***A****=* 0.1937***C*** + 0.0890 ***r*** = 0.9999
where ***A*** is the ratio of peak area and ***C*** is MET concentration in µg mL^−1^, and r is the correlation coefficient.

The low limit of quantitation (LOQ) was calculated for sensitivity, and it was the concentration of analyte that produced a repeatable response that was a minimum five times as the blank. The LOQ calculated was 0.5 µg mL^−1^.

Quality control samples were analyzed five times each, and accuracy was calculated by comparing the averaged measurements to the nominal values and was expressed in % relative error (RE) ((measured concentration − true concentration/true concentration) × 100). The permissible limit was ±15% of the genuine concentration. The computed %RE ranged from −6.82 to 0.99%, according to the values in Table 2. The intraday precision was determined by computing the %RSD for the analysis of the QC samples (n = 5) within one day, whereas the interday precision was determined by analyzing the QC samples on three different days. Table 2 shows that the calculated %RSD values varied from 0.08 to 1.60% (intraday precision) and 0.15 to 4.72% (interday precision), which met the acceptance criteria of the FDA guidelines (±15%). All of these findings demonstrated the new bio-analytical method’s high accuracy and precision.

The chromatograms of blank rat plasma, plasma spiked with target analyte and IS, and plasma samples taken from rats given MET and papaya extract (two hours after administration) and spiked with the prescribed amount of the internal standard were compared to assess specificity. As displayed in Figure 1, the endogenous matrix of plasma had little effect on the isolated peaks, supporting the chromatographic method’s selectivity.

The quality of the extraction process was calculated by comparing the peak area ratios of the analyte in rat plasma at QC concentrations to those in the mobile phase at corresponding concentrations. Table 3 shows that the extraction recovery for MET ranged from 87.56 to 92.22% with a %RSD of 5.32, while the extraction recovery for IS was 90.51% with a %RSD of 3.82. All these results confirmed that the plasma matrix did not affect the extraction efficiency and ensured the efficiency of the extraction method.

The stability of MET in rat plasma was studied for a short period of time (one day) at room temperature (bench top stability), three freeze thaw cycles, and auto-sampler stability (processed sample stability) over a 24 h period. If the medication’s plasma stability results were within 15% of the current concentrations, the medication was pronounced stable. The final results were 108.30% ± 7.38, 95.03% ± 8.04, and 97.15% ± 4.35. Table 4 verifies that the QC samples remained steady in all of the tested conditions.

### 2.3. Pharmacokinetics Study Results

After drug administration without papaya extract (group II) or with it (group III), plasma concentrations of the administered drug were considered using the formerly calculated regression equation and then a mean plasma concentration versus time graph was designed, Figure 2, after which the results were compared and the non-compartmental pharmacokinetic study was then carried out using PKSolver.

Looking at Figure 2, MET was promptly absorbed following oral treatment in both groups II and III, however, it reached its maximum concentration in the circulation within a shorter time when administered together with papaya extract (T_max_ = 15 min in group III compared to one hour in group II). As T_max_ is important for estimating the rate of absorption, onset of action, and to evaluate the efficacy of single-dose drugs used in acute conditions, one can conclude that the combination of MET and papaya extract will be an effective and fast hypoglycemic remedy. Additionally, the amount of drug absorbed increased by 7.33% in the group of rats that received both the drug and the plant extract. This may be attributed to the fact that papaya extract contains saponin, which acts as a surface active agent that increases the solubility of MET blood plasma, resulting in an improved absorption rate and extent of the studied drug. Regarding MET’s half-life (t_1/2_), volume of distribution (Vd), and clearance (Cl), they increased by 11.86, 27.16, and 13.51%, respectively, in group III relative to group II. All the results of the pharmacokinetic study are shown in Table 5. Based on the outcomes of the pharmacokinetic study described above, one can draw a conclusion that combining papaya with MET can have a significant and rapid effect on fasting blood glucose and HbA1c, and this can be a viable complementary alternative for diabetic patients who want to get their blood sugar under control quickly.

### 2.4. Evaluation of the Greenness of the Developed Method

Green analytical chemistry aims to make analytical techniques less harmful to the environment and more human friendly. The use of greenness assessment standards necessitates the use of certain tools. Several methods used for GAC measures have been created thus far, with the Analytical GREEnness calculator (AGREE) [44] being one of the most recently used greenness assessment tools. The GAC is a user-friendly, informative, and sensitive metric tool for assessing analytical operations. The assessment criteria are based on the 12 significance principles, and alternative weights can be applied to them, allowing for some flexibility. Each one of the 12 input variables is converted to a common scale ranging from 0 to 1. The sum of the assessment results for each principle is the final assessment result. The result is a graph that looks like a clock, showing the overall score in the center and a color depiction (with values close to 1 and dark green color indicating that the assessed procedure is greener). The intuitive red-yellow-green color scale reflects the procedure’s performance in each principle, while the weight of each principle is expressed by the width of its associated section. The proposed HPLC method has an AGREE score of 0.6 and a middle color of somewhat light green, Figure 3, indicating that it has a low environmental impact and can be regarded a green method.

For further assessment of the greenness of this method, another greenness metric tool, the RGB additive color model [45], was used. It is based on three primary colors (red, green, and blue), which correlate to the three main characteristics of each analytical method: analytical performance (R), safety/ecological friendliness (G), and productivity/practical effectiveness (B). If an analytical method meets all of the primary criteria to a satisfactory degree, it is classified as white. If it is suitable in terms of only two attributes, the method acquires yellow, magenta, or cyan, and if it has only one attribute and lacks the other two, it acquires red, green, or blue. The evaluation of greenness is objectively quantified by a Color Score (CS) ranging from 0% to 100%, with a CS of 66.6% corresponding to “satisfaction range” and a CS of 33.3% representing “tolerance range”.

This tool also features a quantitative parameter that combines all three scales by applying different weights to them, which are defined subjectively by the user (W). This is called “Method Brilliance” (MB). It refers to a method’s genuine capabilities and expresses its “perfection or flawlessness.”

Based on the RGB results in Table 6, it can be stated that the devised approach has outstanding precision, accuracy, and selectivity (scores of 70, 75, and 80, respectively). In terms of the environment (green color), only minor amounts of harmful chemicals are used, and there is no occupational risk (scores of 70, 75, and 90, respectively). While for the blue aspect, a little amount of raw sample is required (score = 80), the analysis cost and sample overall are both moderate (scores = 66.6 and 50, respectively). As a result, this approach cannot be classified as blue, despite the fact that its superior analytical capabilities and obvious greenness contribute to its ultimate yellow color. It is an indication of a method’s broad application, and it is the one to use if the number of analytes to be measured is not too high. Furthermore, the MB was calculated as 67.3%.

## 3. Materials and Methods

### 3.1. Apparatuses

*For powdering of the plant seeds:* Jet mill machinery (Mill Jet and Mill Classifier, ALPA, 1-1000V, ISO9001, CE, Shandong, China).

*For extract evaporation*: A rotary evaporator (Buchi Rotavapor R-300, Cole-Parmer, Vernon Hills, IL, USA).

Rongtai micropipettes with adaptable volume (100–1000 L) and (0.1–100 L) (Mainland, Shanghai, China).

A VM vortex blender 250 (Hwashin, Seoul, Korea).

80–2C Low-speed Electric Centrifuge (Zjmzym, China), (4000 rpm) and (12 tube × 20 mL).

Dionex Ultimate 3000 UHPLC that has a quaternary solvent transfer pump, diode array detection system, and automatic sampler (Germany). Chromeleon software was used.

A digital balance was also used (Sartorius, German).

### 3.2. Chromatographic Conditions

Separation was performed using a developing system: 0.5 mM of KH_2_PO_4_ solution and methanol (65:35, *v*/*v*), adjusting the flow rate constant at 1.7 mL/min, and scanning at 235 nm. A C_18_ column (250 × 4.6 mm, 5 µm) (ZORBAX Eclipse Plus) (Santa Clara, CA, USA) was used as a stationary phase and the temperature of the column was set unchanged at 25 °C. A total of 30 µL of each sample was injected and GUF was chosen as the internal standard with a run duration of 7.5 min.

For method optimization different columns were tested including: XBridge C_18_ (25 cm × 4.6 mm, 5 µm particle size) (Milford, MA, USA), ZORBAX Eclipse Plus C_18_ (250 × 4.6 mm, 5 µm) (Santa Clara, CA, USA), XBridge C_8_ (25 cm × 4.6 mm, 5 µm particle size) (Milford, MA, USA), and ZORBAX Eclipse Plus C_8_ (250 × 4.6 mm, 5 µm) (Santa Clara, CA, USA).

### 3.3. Materials and Reagents

***Metformin*** (purity of 100.13 ± 0.89, Chemical Industries Development (CID) Co., Giza, Egypt) and ***Guaiphenesin*** (99% purity, Novartis Pharma S.A.E., Cairo, Egypt).

***Methanol*** (HPLC grade) (Fisher Scientific, Loughborough, UK) and KH_2_PO_4_ and ethanol (El-Nasr Pharmaceutical, Chemical Co., Abu Zaabal, Cairo, Egypt).

***Deionized water*** (SEDICO Pharmaceutical Co., 6th of October City, Egypt).

### 3.4. Solutions

***Stock solutions of metformin and guaiphenesin (IS)*** (1 mg/mL) were prepared in methanol. Their working solutions (0.1 mg/mL) were made by diluting the previously produced stock solutions with the mixture of the optimum mobile phase to make 10 mL solutions. Solutions were then reserved at −20 °C until the study was performed.

### 3.5. Animals Used during the Study

Eighteen adult rats, female Wistar albinos, obtained from the Animal House of Nahda University in Beni-Suef (NUB), Egypt, and weighing 200 ± 50 g, were used in this study. The rats were kept in an air-conditioned (25 ± 1 °C), pathogen-controlled animal room in the Nahda University Animal House for two weeks for adaptation and to investigate differences in pharmacokinetic characteristics between the treatment and control groups before being subjected to the experiments. The rats had freely available forage and water from the tap. The experimental animals were handled according to guidelines set forth by the Nahda University Animal House and accepted by the Department of Pharmacology and Toxicology, Faculty of Pharmacy, Nahda University in Beni-Suef (NUB).

### 3.6. Plant Material Extraction

A kilo of plant seeds was bought on the open market (Harraz store, Beni-Suef, Egypt). Dr. Ahmed M. Sayed, Department of Pharmacognosy, NUB, evaluated and identified the acquired seeds, followed by finely powdering the seeds using small jet mill machinery from the laboratory. To yield 60 g of rough extract, the extract was macerated without stirring with 1 to 1.5 L of 70% ethanol over a period of 3 days. For biological investigation, the extract was saved at −4 °C.

### 3.7. Construction of Calibration Curve and Preparation of Quality Control Samples

Variable amounts of MET (2.5–150 µL) were taken from its working standard solution (100 µg mL^−1^) and placed into a series of test tubes previously containing 0.5 mL of blank rat plasma. From the previously produced working solution (100 µg mL^−1^), 250 µL of GUF (IS) was added to each sample, and the volume was subsequently corrected to 2 mL with methanol. One minute’s worth of vortex mixing was followed by 10 min of centrifugation at 4000 rpm to remove the plasma protein. Supernatants were transferred to different novel tubes and dried completely by evaporation for each sample. MET samples with a contaminant concentration of 0.5–30 µg mL^−1^ with 50 µg mL^−1^ of IS were prepared by reconstituting the deposit resulted from separate drying of each sample with the mobile phase mixture in 0.5 mL. Three injections of 30 µL of each were made into the HPLC apparatus, and the chromatographic conditions were then tracked. The peak area ratio (peak area of MET/peak area of IS) was calculated for each sample and then used to derive the regression equation after building a calibration curve. To further validate the suggested approach, three quality control samples (2, 15, and 28 µg mL^−1^) were prepared and analyzed using the optimum separation conditions.

### 3.8. Administration of Drug and Assembly of Plasma Samples 

Rats were randomly allocated into three weight-matched groups, each containing six rats. The first group (I) was kept as a normal control group and received only saline solution. A total of 300 mg/kg of MET solution in saline was given to the second group (II) ([according to a previously published study [47]), and 300 mg/kg of MET and 500 mg/kg of papaya extract solution were given to the third group (III). Blood samples (about 0.5 mL each) were obtained from the retro-orbital plexus of each rat at diverse intervals (0.25, 0.5, 1, 2, 4, 7, and 24 h) in heparinized tubes. Immediately after collection, blood samples were centrifuged at 4000 rpm for a period of 10 min, and the clear plasma layer of each sample was removed and stored at −20 °C in a deep freezer until the test.

### 3.9. Pretreatment of the Collected Plasma Samples

The method developed by Fares et al. [48] was modified to extract the studied drug from the collected plasma samples. Plasma samples (0.5 mL each) were put into a clean centrifuge tube after they had been thawed to room temperature. A total of 250 µL of IS working solution (100 µg mL^−1^) was then added to each sample, and methanol was then used to adjust the volume to 2 mL. One minute of mixing the materials was followed by a ten-minute centrifugation process. Each sample’s supernatant was accurately transferred to a clean test tube, where it was thoroughly evaporated until the tube was nearly empty. The remainder of each sample was reconstituted with 0.5 mL of solvent mixture of the mobile phase. To estimate administered drug concentrations, the created method’s instructions were followed, and then substitution in the regression equation was completed. 

### 3.10. Studying the Pharmacokinetics of the Chosen Drug

It was possible to plot the graph representing the estimated mean plasma concentration in the independently collected plasma samples from groups II and III, MET (300 mg/kg) and MET + papaya extract (300 mg/kg + 500 mg/kg), respectively, at various times versus the corresponding collection time. A study was then performed on non-compartmental pharmacokinetic parameters in Microsoft Excel [49], using the PKSolver (a publicly accessible, menu-driven application).

## 4. Conclusions

For studying the effects of the concurrent administration of papaya extract on MET pharmacokinetics, a unique ecological RP-HPLC approach was developed and refined. MET solubility and bioavailability are both enhanced by papaya saponin content, which acts as an emulsifying agent. The use of papaya extract in conjunction with MET may be an advantageous treatment for effective and fast diabetes control during acute conditions since MET reaches its plasma peak concentration within 15 min, compared to 1 h for MET alone. Concurrent administration of papaya extract with MET may reduce the risk of long-term MET problems. Furthermore, the proposed RP-HPLC method prioritizes being sensitive as well as a green method with minimal risk to the environment, as evidenced by the findings of the greenness assessment. Furthermore, the sample preparation necessitates a simple protein precipitation procedure, hence a short analysis time.

## Figures and Tables

**Figure 1 molecules-27-00375-f001:**
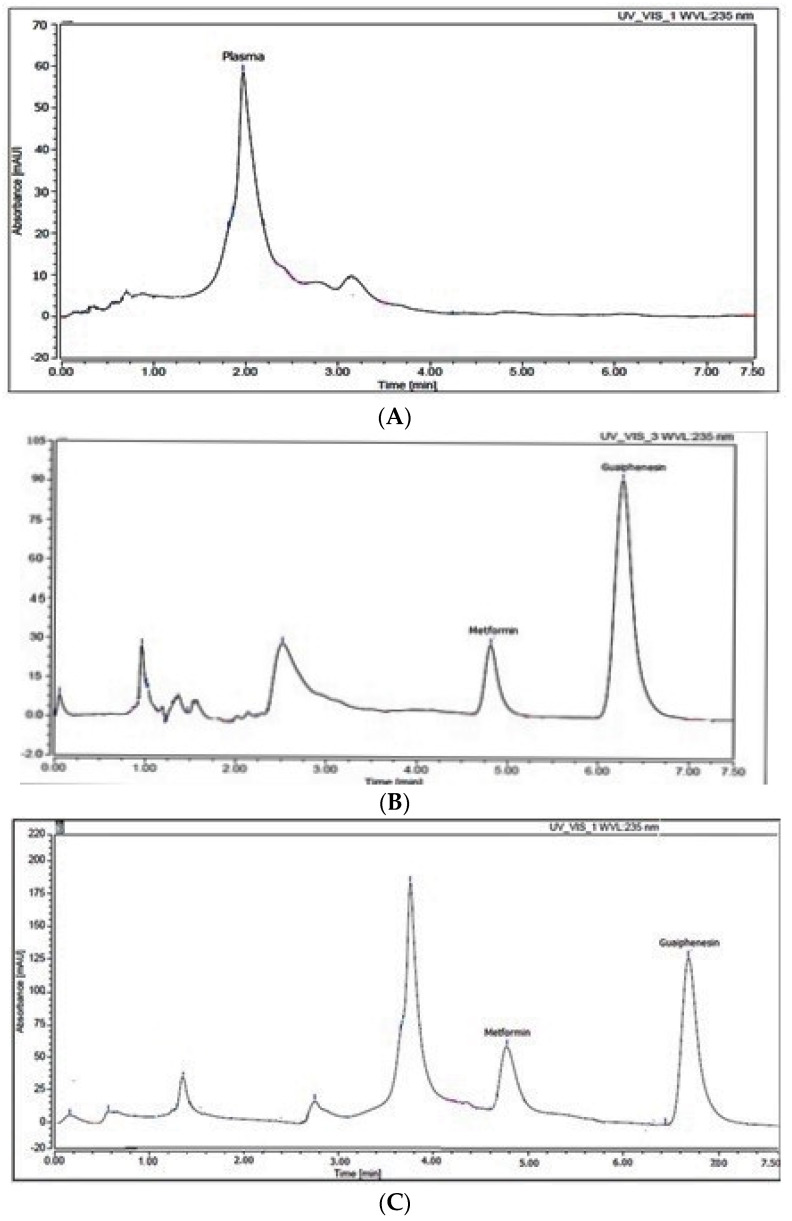
HPLC chromatogram of (**A**): blank plasma, (**B**) plasma spiked with 15 µg mL^−1^ metformin and 50 µg mL^−1^ guaiphenesin, and (**C**) rat plasma sample after 0.25 h (from group III) spiked with 50 µg mL^−1^ guaiphenesin.

**Figure 2 molecules-27-00375-f002:**
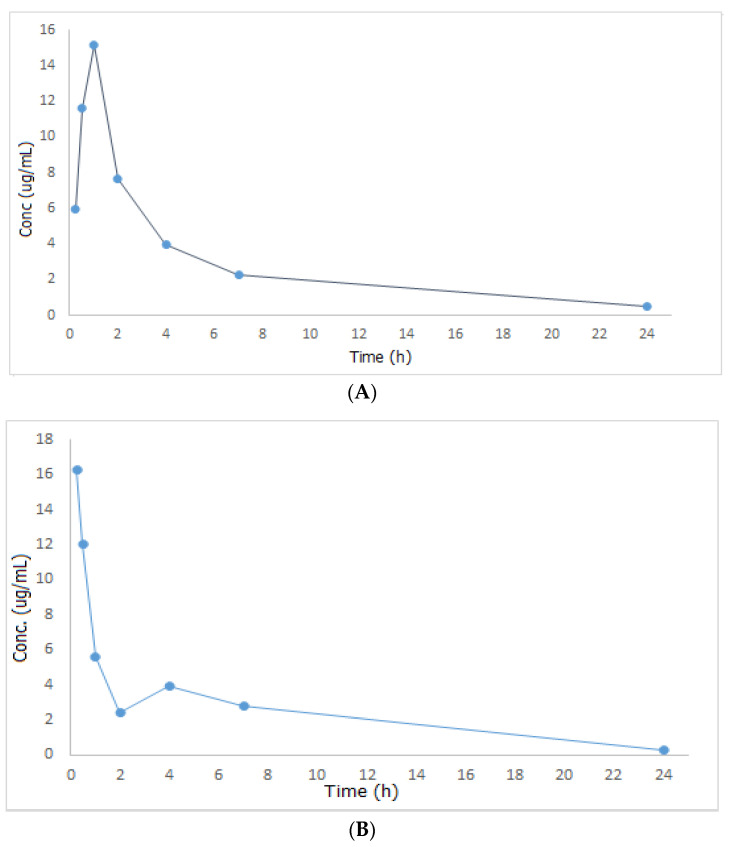
Mean plasma concentration—time curves of metformin after oral administration of 300 mg/kg metformin (**A**) and after oral administration of 300 mg/kg metformin + 500 mg/kg papaya extract (**B**).

**Figure 3 molecules-27-00375-f003:**
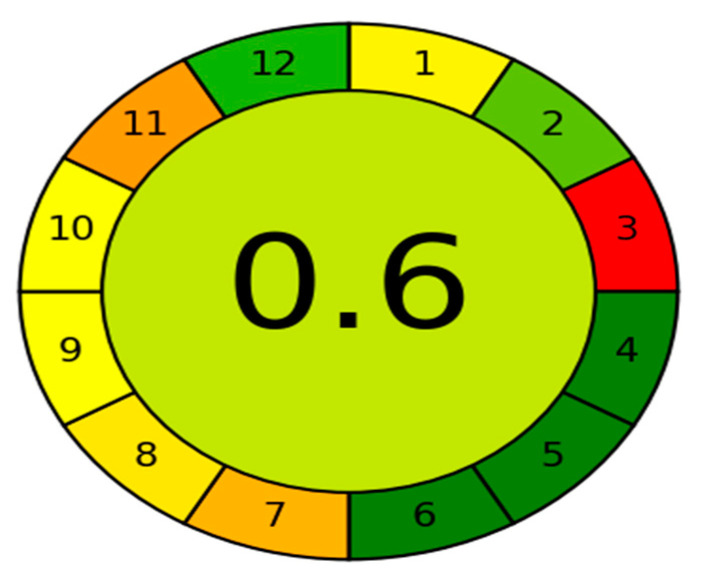
Results of AGREE analysis for the developed HPLC method.

**Table 1 molecules-27-00375-t001:** Extraction recoveries of different tested solvents.

Solvent	Acetonitrile	Ethanol	Methanol	Aqueous Perchloric Acid
**Extraction recovery**	78.45 ± 7.75	82.10 ± 6.74	87.56 ± 5.32	80.98 ± 2.98

**Table 2 molecules-27-00375-t002:** Intra and inter assay precision and accuracy of the proposed method.

Concentration (µg mL^−1^) *	Intraday			Interday		
	Recovery %	%RSD	%RE **	Recovery %	%RSD	%RE **
2.00 (LQC)	100.99	0.27	0.99	100.30	0.57	0.30
15.00 (MQC)	93.18	1.60	−6.82	94.44	4.72	−5.56
28.00 (HQC)	99.30	0.08	−0.70	99.30	0.15	−0.70

* Average of 5 experiments. ** Relative error (RE) = ((measured concentration − nominal concentration)/nominal concentration) × 100.

**Table 3 molecules-27-00375-t003:** Extraction recovery results of the studied drugs in spiked human plasma.

	Concentration of the Analyte (µg mL^−1^)	Recovery % *
	2.00	87.56
	15.00	82.90
	28.00	92.22
**Mean ± %RSD**		87.56 ± 5.32
**IS**	50.00	90.51 ± 3.82

* Average of 5 determinations.

**Table 4 molecules-27-00375-t004:** Stability results of the studied drugs in spiked human plasma at different conditions.

	Recovery % *			
	Concentration of the Analyte (µg mL^−1^)	Three Freeze Thaw Cycles	Bench Top Stability	Short Term Stability for 24 h
	2.00	86.48	92.27	99.13
	15.00	101.17	99.36	113.77
	28.00	97.44	99.82	112.00
**Mean ± %RSD**		95.03 ± 8.04	97.15 ± 4.35	108.30 ± 7.38

* Average of 5 determinations.

**Table 5 molecules-27-00375-t005:** Pharmacokinetic parameters of the developed method.

Parameter	Unit	Metformin (300 mg/kg)	Metformin + Papaya Extract 300 + 500 (mg/kg)
**T_1/2_**	h	2.95	3.30
**T_max_**	h	1.00	0.25
**C_max_**	µg mL^−1^	15.14	16.25
**AUC_0_-**	µg mL^−1^ h	42.82	33.01
**AUC_0_-_inf_**	µg mL^−1^ h	52.60	46.34
**Mean residence time (MRT)**	h	4.01	5.18
**Volume of distribution (Vd/F)**	L	24.25	30.83
**Clearance (Cl/F)**	L/h	5.70	6.47

**Table 6 molecules-27-00375-t006:** Evaluation of the developed HPLC method using RBG additive color model.

			Accuracy (Bias%)		Precision (%RSD)		Extraction Efficiency		Sensitivity		Selectivity	
**REDNESS (analytical performance)**		W = 1										
CS:	67.1%	LAV = 33.3	15	15	15	15	75	75	L	L	L	L
		LSV = 66.6	3	3	3	3	90	90	H	H	H	H
		Result	2.51	2.51	1.23	1.23	87.5	87.5	M	M	VH	VH
		Score (0–100)	**70**	**70**	**75**	**75**	**65**	**65**	**50**	**50**	**80**	**80**
			**Amount of chemicals**		**Toxicity**		**Waste**		**Occupational hazards**		**Energy**	
**GREENNESS (safety and eco-friendliness)**		W = 1										
CS:	70.4%	LAV = 33.3	50	50	10 pictograms in total	10 pictograms in total	20	20	H	H	1.5	1.5
		LSV = 66.6	15	15	5 pictograms in total	5 pictograms in total	10	10	L	L	<1.5	<1.5
		Result	12.75	12.75	3 pictograms in total	3 pictograms in total	12.75	12.75	no	no	<1.5	<1.5
		Score (0–100)	**70**	**70**	**75**	**75**	**55**	**55**	**90**	**90**	**66.6**	**66.6**
			**Analysis cost**				**Sample throughout**			**Sample material consumption**		
**BLUENESS (productivity/practical effectiveness)**		W = 1										
CS:	64.6%	LAV = 33.3	H	H	H	H	5 samples/h	5 samples/h	5 samples/h	H	H	H
		LSV = 66.6	M	M	M	M	15 samples/h	15 samples/h	15 samples/h	M	M	M
		Result	M	M	M	M	8 sample/h	8 sample/h	8 sample/h	VL	VL	VL
		Score (0–100)	**66.66**	**66.66**	**66.66**	**66.66**	**50**	**50**	**50**	**80**	**80**	**80**
**FINAL COLOR:**			**REDNESS**		**GREENNESS**		**BLUENESS**		**BRILLIANCE (MB):**		**67.3%**	
**YELLOW**			**≥33.3%**	**≥66.6%**	**≥33.3%**	**≥66.6%**	**≥33.3%**	**≥66.6%**				
			**yes**	**yes**	**yes**	**yes**	**yes**	**no**				
**Short annotation: 67.3 yellow**			**Long annotation: 67.3 yellow (67.1/1 red − 70.4/1 green − 64.6/1 blue)**									

Very low (VL); low (L); high (H); moderate (M).

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
