# Peer review of "Development and Greenness Assessment of HPLC Method for Studying the Pharmacokinetics of Co-Administered Metformin and Papaya Extract"

_molecules, 2022, doi:10.3390/molecules27020375_

Round 1

Reviewer 1 Report

Dear Authors,

An interesting work on the Development and Greenness Assessment of HPLC Method for 2 Studying the Pharmacokinetics of Co-administered Metformin 3 and Papaya Extract. The combination of papaya and MET significantly changed the pharmacokinetics of the 40 drugs. As a result, this combination will be very beneficial in maintaining rapid blood 41 glucose balance. The manuscript was written well without grammatical errors. In the article, the methods are clearly stated and organized logically.

I have a few comments. Table 5, which analyzes the suggested HPLC approach based on the RBG additive color model, might be simplified and made more comprehensible (Line 292)

Author Response

RBG is calculated using a reference excel sheet, so table 6 (table 5 previously) was taken directly from the original soft data sheet. However, the table is adjusted to be more clear with highlighting its illustration paragraph.

Reviewer 2 Report

The work hereby presented the development and greenness assessment of an HPLC method and its application in pharmacokinetic studies of Co-administered metformin and papaya extract.

The work is well-organized and well-presented. The literature review is explained in detail and the references are comprehensive and appropriate. The experimental part is well organized and described in detail. In general, the herein reported study is interesting and novel. I therefore recommend publication after some corrections.

Line 34: Delete the comma after the word model.

Line 151: Please provide more details regarding the optimization of the sample preparation procedure. Also please include a table with the extraction recoveries that were found for each protein precipitation reagent.

Line 179: Please replace “chosen” with “calculated”

Section 3.1 and 3.2: The authors repeat the information regarding the column, the mobile phase etc. in both sections. Please provide the information regarding the chromatographic system only at section 3.2.

Line 355: Please replace “μl” with “μL”

Author Response

Response to Reviewer 2

Line 34: Delete the comma after the word model.

done

Line 151: Please provide more details regarding the optimization of the sample preparation procedure. Also please include a table with the extraction recoveries that were found for each protein precipitation reagent.

The required optimization details of sample preparation procedure was highlighted in line 369.

A table with the extraction recoveries was added as table 1

Line 179: Please replace “chosen” with “calculated”

Done and highlighted

Section 3.1 and 3.2: The authors repeat the information regarding the column, the mobile phase etc. in both sections. Please provide the information regarding the chromatographic system only at section 3.2.

Done.

Line 355: Please replace “μl” with “μL”

Done

Reviewer 3 Report

The manuscript entitled Development and greenness assessment of HPLC method for studying the pharmacokinetics of co-administered Metformin and Papaya extract. The manuscript needs substantial revision; several comments need to be addressed before getting it considered for publication. 

  1. Abstract: please remove the methods, results, and conclusion words.
  2. Line 58: Carica papaya should be italic.
  3. Please, give in the Introduction more scientifically sound explanation how it did happen that so different activities had been chosen for investigation, giving their different mechanism of action
  4. For all tested activities, please explain the chosen doses for experiments
  5. Line 254: please rewrite
  6. Citations should be provided for all methods used in the study. 
  7. Statistical analysis was missing.
  8. Please, give the protocol number of Committee approval for performing the animal studies.
  9. Please, give scientifically sound explanation of how the doses in each biological activity testing were chosen? Why only two different doses were chosen for most of the experiments?
  10.  How to measure the  HbA1c  level?
  11. Are the RGB results of precision, accuracy, and selectivity measured in the repetitive way? Deduce it.
  12. Which type of mobile phase was used. If it is polar or nonpolar base; why? 
  13. If the authors add GC-MS of Carica papaya extract, it helps to correlate compound with activity and know about how to reduce the risk of long-term MET problems.
  14. Line 204: please rewrite.
  15. Is it showed any toxicity about co-administration of drugs, If yes, how to overcome it.

Author Response

Response to Reviewer 3

  1. Abstract: please remove the methods, results, and conclusion words.

done

2.     Line 58: Carica papaya should be italic

done

3.     Please, give in the Introduction more scientifically sound explanation how it did happen that so different activities had been chosen for investigation, giving their different mechanism of action

The required explanation of different papaya activities was added in the introduction section and highlighted.

  1. For all tested activities, please explain the chosen doses for experiments

The dose of metformin (300 mg/Kg) was chosen according to a previously published work and cited within the manuscript. Regarding papaya dose (500 mg/Kg), by testing LD50, it showed no toxicity up to 5g/Kg. So, dose of 500 mg/Kg was chosen for performing this study.

  1. Line 254: please rewrite

The sentence was omitted from the manuscript

6.     Citations should be provided for all methods used in the study. 

Done

  1. Statistical analysis was missing.

No published method for studying the pharmacokinetics of metformin when taken with papaya extract to be compared with.

  1. Please, give the protocol number of Committee approval for performing the animal studies.

Done

  1. Please, give scientifically sound explanation of how the doses in each biological activity testing were chosen? Why only two different doses were chosen for most of the experiments?

The dose of metformin (300 mg/Kg) was chosen according to a previously published work and cited within the manuscript. Regarding papaya dose (500 mg/Kg), by testing LD50, it showed no toxicity up to 5g/Kg. So, dose of 500 mg/Kg was chosen for performing this study.

The main target of this work is to study the pharmacokinetic parameters of metformin and so, only one dose for each of metformin and papaya extract was sufficient for achieving our goal.

  1. How to measure the  HbA1c  level?

The authors did not measure HbA1c but metformin effect on HbA1c was expected from the pharmacokinetics results.

  1. Are the RGB results of precision, accuracy, and selectivity measured in the repetitive way? Deduce it.

It was performed 5 times and this was mentioned under table 3 (table 2 previously)

  1. Which type of mobile phase was used. If it is polar or nonpolar base; why? 

The used mobile phase was relatively polar.

  1. If the authors add GC-MS of Carica papaya extract, it helps to correlate compound with activity and know about how to reduce the risk of long-term MET problems.

GC-Mass is only used for volatile constituents but are targeting the whole constituents of papaya.

  1. Line 204: please rewrite.

done

  1. Is it showed any toxicity about co-administration of drugs, If yes, how to overcome it

No toxicity was observed  about co-administration of drugs upon performing the rat model for one month.

Round 2

Reviewer 3 Report

The manuscript is suitable to publish in the present form.